# Role of IL-4 and IL-13 in Cutaneous T Cell Lymphoma

**DOI:** 10.3390/life14020245

**Published:** 2024-02-09

**Authors:** Roberto Mazzetto, Paola Miceli, Jacopo Tartaglia, Christian Ciolfi, Alvise Sernicola, Mauro Alaibac

**Affiliations:** Dermatology Unit, Department of Medicine (DIMED), University of Padua, 35121 Padova, Italy; robertomazzetto93@gmail.com (R.M.); paolamiceli9@gmail.com (P.M.); jacopo.tartaglia@studenti.unipd.it (J.T.); chriciolfi96@gmail.com (C.C.); mauro.alaibac@unipd.it (M.A.)

**Keywords:** interleukin 4, interleukin 13, cutaneous T cell lymphoma, mycosis fungoides, Sezary syndrome, dupilumab, nemolizumab, tralokinumab, lebrikizumab, tezepelumab

## Abstract

The interleukins IL-4 and IL-13 are increasingly recognized contributors to the pathogenesis of cutaneous T cell lymphomas (CTCLs), and their role in disease-associated pruritus is accepted. The prevailing Th2 profile in advanced CTCL underscores the significance of understanding IL-4/IL-13 expression dynamics from the early stages of disease, as a shift from Th1 to Th2 may explain CTCL progression. Targeted agents blocking key cytokines of type 2 immunity are established therapeutics in atopic disorders and have a promising therapeutic potential in CTCL, given their involvement in cutaneous symptoms and their contribution to the pathogenesis of disease. IL-4, IL-13, and IL-31 are implicated in pruritus, offering therapeutic targets with dupilumab, tralokinumab, lebrikizumab, and nemolizumab. This review analyzes current knowledge on the IL-4/IL-13 axis in mycosis fungoides and Sezary syndrome, the most common types of CTCL, examining existing literature on the pathogenetic implications with a focus on investigational treatments. Clinical trials and case reports are required to shed light on novel uses of medications in various diseases, and ongoing research into the role of IL-4/IL-13 axis blockers in CTCL therapy might not only improve the management of disease-related pruritus but also provide in-depth insights on the pathophysiologic mechanisms of CTCL.

## 1. Introduction: Cutaneous T Cell Lymphoma

Cutaneous T cell lymphomas (CTCLs) are a heterogeneous group of lymphoproliferative disorders derived from skin-homing effector memory T cells [1]. Typically exhibiting a male predominance, CTCL manifests in the fifth and sixth decades of life [2].

Mycosis fungoides (MF) is the most common subtype that accounts for approximately 60% of CTCL [3]. Skin lesions typically present with erythematous and scaly patches in early stage but gradually form generalized plaques, tumors, or erythroderma in the advanced stage [4]. The appearance of erythema is a potential source of severe esthetic impairment and a substantial concern for patients irrespective of subjective symptoms of pruritus [5]. Sezary syndrome (SS) is an aggressive CTCL variant that is characterized by peripheral blood lymphocytosis, which commonly presents with widespread erythema [4].

Recent studies shed light on the origin of malignant T cells, revealing distinct phenotypic characteristics. SS is associated with the neoplastic evolution of central memory T cells, expressing C-C chemokine receptor type (CCR) 7, CCR4, L-selectin, and CD27, while MF is linked to skin-resident effector memory T cells, lacking CCR7/L-selectin and CD27 expression, but featuring high levels of CCR4 and cutaneous lymphocyte antigen (CLA) [6,7]. Disease progression correlates with a decrease in the T cell receptor (TCR) repertoire, emphasizing the clonal nature of malignant CD4+ T cell populations [8]. Aberrant cytokine expression in the MF/SS tumor microenvironment (TME) is one of the most important factors in disease pathogenesis and progression [9,10]: while reactive T helper type 1 (Th1) and CD8+ tumor-infiltrating lymphocytes are found in the TME of early-stage MF/SS, disease progression is accompanied by infiltration with benign and malignant T lymphocytes producing mostly T helper type 2 (Th2) cytokines (IL-4, IL-5 and IL-13) [11]. Within patches, plaques, and tumoral CTCL lesions, nonneoplastic T cells such as macrophages, fibroblasts, dendritic cells, mast cells, myeloid-derived suppressors and other cells comprise the TME. An increasing amount of evidence suggests that the TME plays important roles in the tumor occurrence and development of MF and SS, both of which are regarded as Th2-dominant diseases [12]. The cytokine production model is considered of great importance for the TME of CTCL and its role can be divided into two aspects: one is to support tumor growth and survival, and the other is to suppress antitumoral immune response causing an inability to reject tumors and susceptibility to infection [13,14].

The systemic immune response is critically dependent on coordinated lymphocyte migration and recirculation. This “homing” guides naïve memory T lymphocytes to the microenvironments that control their differentiation and survival, and target effector lymphocytes to sites of antigenic insult [15]. In case of malignant transformation, however, this intrinsic behavior forms a serious threat to the organism, because it permits rapid tumor dissemination and early spread in most types of cancer. However, unlike the metastatic spread of other cancers, lymphoma dissemination generally is a reflection not of tumor progression but of a conserved physiological behavior [15]. The immunological and clinical characteristics of these lymphomas are thus closely linked to the physiological function of the lymphocytes from which they originate in the skin (effector and central memory T cells in MF and SS, respectively).

Atopic dermatitis (AD) has been associated with immunological dysfunction which may influence the risk of cancer development; in particular, severe AD was associated with up to threefold higher risk of lymphoma [16]. In particular, four hypotheses have been proposed to account for this observed relationship: chronic inflammation, immunosurveillance, exposure to immunomodulating drugs, and inappropriate Th2-immune skewing [17]. These aspects could thus justify the potential utility of therapies for dermatological inflammatory conditions, such as AD, in the setting of lymphomas.

This review analyzes current knowledge on the IL-4/IL-13 axis in mycosis fungoides and Sezary syndrome, the most common types of CTCL, examining existing literature on the pathogenetic implications with a focus on investigational treatments. To achieve this objective, PubMed was searched from inception to December 2023 using the terms “cutaneous lymphoma” and “IL-4” or “IL-13” or “IL-31” or “TSLP” or “OX-40”. Papers written in the English language were screened by the authors to retrieve results relevant to the topic of this review; moreover, the reference list of included papers was analyzed to identify additional results.

## 2. IL-4 and IL-13 in the Development and Progression of Cutaneous T Cell Lymphoma

Large-scale genome studies and advances in sequencing technologies have revealed a comprehensive landscape of genomic and epigenetic modifications that constitute crucial pro-tumorigenic factors in the initiation and progression of CTCL [18]. Recent studies have illuminated the dysregulation of various signaling pathways following genomic events. These include the Janus kinase/signal transducer and activator of transcription proteins (JAK/STAT), nuclear factor kappa-light-chain-enhancer of activated B cells (NF-κB), Phosphoinositide 3-kinases/mammalian target of rapamycin (PI3K/mTOR), and mitogen-activated protein kinase (MAPK) pathways, which collectively contribute to the lymphomagenesis and heterogeneity observed in T cell malignancies [19].

In early stage CTCL, upregulation of STAT4, an important transcription factor for the Th1 lymphocyte subset, is observed. Several dysregulated genes have been identified across multiple studies in addition to STAT4, including Plastin-3 (PLS3), killer cell immunoglobulin-like receptor (KIR) 3DL2, and twist-related protein 1 (TWIST1) [20]. However, as the disease advances, STAT4 expression is often lost contributing to a switch from Th1 to Th2 and correlating clinically with worse prognosis [21]. During this transition, STAT6 expression is frequently upregulated in CTCL as stages advance [21]. Simultaneously, the activation of STAT3 and STAT5 may shift to a cytokine-independent state, driven by constitutive activation of JAK1 and JAK3 kinases [21]. STAT6 belongs to a family of seven similar members and it is primarily stimulated by IL-4 and IL-13, acting as a Th2-inducing factor [22]. It is implicated in the pathophysiology of various allergic conditions, such as asthma, AD, eosinophilic esophagitis, and food allergies, but also in TME regulation [22]. It is also noteworthy that CCR4, a biomarker associated with CTCL and AD, is induced by binding of the C-C motif chemokine ligand (CCL17), which is upregulated in Th2 skin diseases [23]. CCR4 acts as a prominent biomarker expressed in the skin lesions of MF and its inhibition by monoclonal antibody mogamulizumab has proven to be an effective therapy for advanced MF [24,25].

MF and SS are frequently associated with eosinophilia and high IgE levels [26], that contribute to a Th2-skewed TME supporting further immunosuppression and heightened susceptibility to S. aureus infections, in addition to worsened itch and skin involvement [26]. Periostin, an extracellular matrix protein produced by fibroblasts, induces chronic inflammation by stimulating thymic stromal lymphopoietin (TSLP) production. TSLP directly stimulates keratinocyte activation in AD and tumor cell growth in CTCL, inducing a Th2-dominant tumor environment [27]. Moreover, Human leukocyte antigen G (HLA-G), a nonclassical class Ib molecule, can be induced by IL-10, an interleukin associated with the expression of a Th2 cytokine profile [28]. This association has been observed in various primary cutaneous lymphomas (CL), impacting the effectiveness of the tumor-associated immune response in CL and facilitating the transition from low- to high-grade lymphoma [28].

Furthermore, there is an ongoing debate about the presence of a direct relationship between chronic dermatoses, atopy, and subsequent CTCL development [29]. Bacterial skin infections are a common cause of morbidity in patients with CTCL, and represent leading causes of morbidity and mortality [13]. CTCL lesions, compared with psoriatic lesions that typically display enhanced antimicrobic responses, presented an impaired regulation of antibacterial proteins (ABPs), with levels even below those found in AD. This aspect was associated with a relative deficiency in the ABP-inducing cytokine IL-17 and a strong presence of the ABP-downregulating cytokine IL-13 [30]. IL-31 is a recently identified cytokine with a well-defined role in the pathogenesis of pruritus and is produced predominantly by circulating Th2 lymphocytes [31]. Increased IL-31 serum levels were detected in patients with CTCL, compared to healthy controls, and in patients with advanced-stage CTCL, compared to those with early-stage disease [32]. Type 2 inhibition may therefore improve disease control, at least symptomatically, considering its impressive importance in MF and SS [33]. Bexarotene and extracorporeal photochemotherapy, two main therapies adopted in the management of CTCL, have for example the capacity to inhibit IL-4 production in the peripheral blood cells of SS patients [34,35]. Therefore, it could be also plausible that IL-4 and/or IL-13 inhibitors as well as IL-31 inhibitors could be not only safe but also beneficial for patients with CTCL [36] (Figure 1).

## 3. Targeting IL-4 and IL-13 in Cutaneous T Cell Lymphoma

Dupilumab is a fully human monoclonal antibody that binds the IL-4 receptor (IL-4R) alpha subunit and inhibits signaling of IL-4 and IL-13. It was approved by the Food and Drug Administration (FDA) for the treatment of moderate-to-severe AD, of which itching is a significant symptom. Many disorders, both cutaneous and extracutaneous, can cause pruritus, which is also a substantial symptom of hematologic malignancies or solid tumors and is notoriously difficult to treat. Pruritus can impair sleep quality and work productivity, leading to depression and suicide ideation; therefore, recent studies aimed to explore the clinical and immunological effects of type 2 cytokine blockade with dupilumab as supportive treatment for pruritus across a variety of clinical conditions.

Among immune cell populations, Th2 lymphocytes and mast cells are key players in the physiology of itching secondary to an inflammatory process; the cytokines IL-4, IL-13, and IL-31 play major roles in the itching process. Therapeutic blocking of the IL-4 and IL-13 receptors with dupilumab in CTCL is thought to reduce the synthesis of IL-31, the neuroimmune ‘‘itch cytokine’’, alleviating cancer-related pruritus by directly affecting neurons, even in cases unrelated to Th2 cell inflammation. A recent case study published by Talmon et al. in 2023 presents the initial evidence of the tolerability and response to dupilumab in intractable malignancy-associated pruritus in three patients [37]. None of the three patients showed clinical evidence of AD or other causes of itching (e.g., uremia or liver failure), and none responded to conventional treatments for pruritus, such as corticosteroids and oral anti-H1 antihistamines. Biweekly administration of dupilumab led to a rapid improvement in itching and the symptom subsided entirely after a few doses without any significant adverse effects. These encouraging clinical observations provide a basis for further investigations in a larger cohort: the efficiency of dupilumab for treating cancer-associated pruritus should be tested in a large randomized clinical trial.

SS is a leukemic variant of CTCL, characterized by erythroderma, lymphadenopathy, and peripheral blood involvement. In addition to a poor prognosis, SS patients suffer from intense pruritus causing a profound reduction in the quality of life. Th2 cytokines are key drivers of pruritus in SS via multiple mechanisms; thus, targeting Th2 cytokines in SS is a promising approach to revert immunosuppression by promoting Th1 responses and to improve supportive care by reducing itching. In a case report, Steck et al. treated with dupilumab in combination with continued extracorporeal photopheresis a SS patient with stable disease but intractable pruritus [38]. The patient had been suffering with pruritus for two years, and previous treatments with narrowband UVB, topical and systemic steroids, and cyclosporine A failed to control the disease. Given the patient’s intractable pruritus and their refusal to escalate systemic treatment, off-label supportive dupilumab treatment over a period of 44 weeks was administered (600 mg s.c., followed by 300 mg s.c. biweekly). Clinical and immunological monitoring on blood and skin samples from the patient was performed over 44 weeks, and in vitro assays with the patient’s lymphoma cells were performed to address the effects of dupilumab on the Sézary cells’ response to Th2 cytokines. After initiation of treatment, skin disease, pruritus, and QoL markedly improved. Intriguingly, in both blood and skin, a reduction in Th2 bias was observed. Moreover, lymphocyte counts and Sézary cells in blood increased and later stabilized under dupilumab treatment. In vitro assays showed that dupilumab abrogated the anti-apoptotic and activating effects of Th2 cytokines on Sézary cells. These data support the future exploration of Th2 modulation as an adjunctive management strategy in SS. Although dupilumab is well tolerated and effective for the treatment of atopic and allergic conditions, and potentially for treating cancer-associated pruritus, clinicians should consider its potential for unexpected adverse events. The current knowledge on the safety profile of dupilumab was reviewed in 2022 [39]: injection-site reactions are the most commonly reported event. Dupilumab is notably related to ophthalmic complications (e.g., dry eyes, conjunctivitis, blepharitis, keratitis, and ocular pruritus); therefore, an ophthalmological examination for the presence of potential predictive indicators of ophthalmic adverse events is recommended before initiation of therapy. A recent prospective study in 2023 assessed potential predictive biomarkers in the tear fluid of 39 AD patients initiating dupilumab: ocular adverse events during treatment were significantly associated with low tear break-up time and with high IL-33 levels [40]. Other adverse events include paradoxical head and neck dermatitis, onset of psoriatic lesions, alopecia areata, hypereosinophilia, arthritis, and reportedly CTCL. Most of these conditions can be managed while continuing dupilumab treatment but some may result in a discontinuation of treatment. Their origin is still incompletely understood (such as that of severe conjunctivitis) and requires further investigation, although it has been suggested that Th2 inhibition may worsen Th1/Th17-dependent immune responses.

As far as the risk of CTCL is concerned, Tran et al. presented the first case of SS developing following the administration of dupilumab in 2020 [41]. The patient experienced a continued progression of AD despite the use of topical corticosteroids and ultraviolet light therapy. In June 2019, the patient was initiated on subcutaneous dupilumab, but two weeks after the treatment they showed persistent erythroderma and lymphadenopathy. Flow cytometry showed a markedly increased CD4 to CD8 ratio of 43:1. Atypical CD4+ T lymphocytes were significantly increased in number, demonstrating partial loss of CD7 expression. Moreover, TCR gene rearrangement studies were performed, revealing monoclonal TCRgamma gene rearrangements and polyclonal TCRbeta gene rearrangements in both the skin and blood. These immunophenotypic findings in the setting of erythroderma led to a SS diagnosis. A logical consequence of dupilumab’s effect of inhibiting Th2 responses would be to counter the progression of SS via a reduction of atypical lymphocytic proliferation; therefore, the development of SS in this patient was unexpected, but the temporal relationship between the initiation of dupilumab and the onset of erythroderma suggests that dupilumab was a trigger for SS in this case.

Since treating AD with dupilumab may be associated with the progression of underlying MF [42,43], a cross-sectional study by Hamp et al. aimed to examine the associations between length of dupilumab treatment, age and sex, and onset of MF [44]. The results suggest a correlation between the duration of dupilumab treatment and the diagnosis of MF; in particular, a higher MF stage at diagnosis was related to a shorter time from dupilumab initiation to MF onset. In addition, elderly male patients seemed to be more at risk as both male sex and older age correlated with the hazard of MF diagnosis. Further investigation on the relationship between dupilumab and MF can shed more light on the question as to whether dupilumab unmasked preexistent MF that had been misdiagnosed as AD in these patients or whether MF truly is an adverse effect of treatment with dupilumab.

Finally, in addition to the association of CTCLs with dupilumab, Nakazaki et al. reported the first case of Hodgkin lymphoma (HL) in a patient treated with dupilumab for one year, who was diagnosed with a rare combination of discordant lymphomas of HL and peripheral T cell lymphoma, based on multiple biopsies [45]. These findings demonstrate a need to be alert for the potential development of lymphomas associated with the IL-13 and IL-4 pathways in patients with poorly responsive AD receiving dupilumab. As both discordant lymphomas of HL and peripheral T cell lymphoma are known to overexpress IL-13, future studies should evaluate the effect of anti-IL-13 therapy in these settings.

## 4. Targeting IL-13 in Cutaneous T Cell Lymphoma

IL-13 is a recognized contributor to the proliferation of certain solid and hematologic malignancies that express its receptor IL-13Ralpha1—such as chronic lymphocytic leukemia and HL [46,47,48]—as well as the decoy receptor IL-13Ralpha2 [49]. Additionally, IL-13 supplies the TME with immunosuppressive signals that hamper anti-cancer immunosurveillance [50]. An IL-13Ralpha2-targeted cytotoxin (IL-13-PE38) has been investigated in clinical trials of patients with brain tumors and in mouse models of oral squamous cell carcinoma highlighting that IL-13Ralpha2 expression may have a therapeutic significance [51]. While T cells physiologically do not express receptors for IL-13 [52], the latter have been detected in high concentrations in malignant T cells from cutaneous localizations of CTCL as well as from the bloodstream of SS patients [53] supporting a role of IL-13 receptors as biomarkers of malignancy throughout each stage of CTCL. Therefore, IL-13 produced by malignant lymphocytes of CTCL provides an autocrine proliferative signal to these cells through activation of the transcription factor STAT6. Furthermore, experimental blocking of the IL-13 signaling in vitro, with an anti-IL-13 antibody or with soluble IL-13Ralpha2, demonstrated a potent antiproliferative effect while viability of the malignant cells remained substantially unchanged [53]. Also, blocking STAT6 downstream of IL-13 signaling—and of that of IL-4—achieved a similar effect in vitro, but it must be considered that transcription factors are generally less druggable intracellular targets compared to extracellular cytokines and their surface receptors, while the effects of JAK inhibition are broader and outside the scope of this review [54]. In this context, a prospective approach targeting IL-13 may work synergistically with current CTCL drugs with a direct anti-tumor effect. Moreover, blocking IL-13—alone or in conjunction with IL-4—may restore antitumor immunity in the local microenvironment by lessening Th2 skewing which is associated with immune escape in CTCL [55]. Finally, targeting IL-13 may also be beneficial to counter its profibrotic effects on dermal fibroblasts that may be responsible for the fibrosis observed in MF [56].

Two anti-IL-13 monoclonal antibodies are currently available in the therapeutic armamentarium for dermatologists. Tralokinumab is a tolerated and effective treatment for moderate-to-severe AD in patients aged above 12 years that was approved in 2021 by the European Medicines Agency (EMA), and subsequently by the FDA, at the dose of 600 mg subcutaneously followed by 300 mg every two weeks. Similarly, lebrikizumab was approved in 2023 by the EMA with an equivalent indication at the dose of 500 mg subcutaneously at baseline and after 2 weeks, followed by 250 mg every two weeks. Both treatments allow for a maintenance dose once every four weeks after clinical response has been achieved. Finally, it has been established that neither tralokinumab nor lebrikizumab interfere with the binding and regulation by IL-13Ralpha2 decoy receptor [57]. Considering the mechanisms of action of these agents and the prominent involvement of IL-13 in CTCL pathogenesis, further exploration of their potential in CTCL treatment is warranted.

## 5. Targeting IL-31 in Cutaneous T Cell Lymphoma

The management of pruritus constitutes a largely unmet clinical need in the current management of CTCL, considering that this symptom extensively affects the daily life of patients and that it is not always controlled by lymphoma-directed treatments. IL-31 is a key player in the pathogenesis of pruritus, both in the acute and in the chronic stages [58], and its signaling pathway is closely regulated by IL-4 and IL-13 [59].

CD4+ skin-directed T lymphocytes, together with cells of innate immunity such as ILC2, are the primary source of IL-31. This cytokine binds a two-subunit receptor, IL-31 receptor A and oncostatin M receptor beta, and activates intracellular signaling mediated by JAK and STAT3 [60,61]. Additional cell pathways involved include PI3K/Akt and MAPK; overall, IL-31 produces effects on cutaneous barrier function, hematopoiesis and immune system regulation [58]. Key mediators of type 2 immunity, IL-4 and IL-13, can upregulate the IL-31 receptor escalating itching responses across conditions that are characterized by Th2 skewing, including CTCL. Moreover, in the cutaneous inflammatory microenvironment, IL-31 is involved in the intense crosstalk between cytokines and chemokines: among the latter, CCL17 and CCL22 are induced by IL-31 and provide ligands for the CCR4 receptor, which is highly expressed on both benign and malignant CD4+ skin-homing T cells [59]. Chemokine-mediated recruitment, together with the release of alarmins such as IL-33 from keratinocytes in response to scratching, boost IL-31 responses and underly a vicious itch–scratch cycle [62]. In the setting of lymphoproliferative disorders, involvement of IL-31 has been established in HL, of which generalized pruritus is a hallmark symptom, in follicular B cell lymphoma, which is outside the scope of our current analysis, and in CTCLs, which mostly comprise malignant CD4+ skin-homing T cells and show skewing towards Th2 [63]. However, the extent of the correlation between IL-31 and pruritus is still debated in this setting and a broader role of this cytokine in the pathogenesis of CTCL has been inconsistently confirmed in the current literature [64].

Moreover, subjects with CTCL and itching show high IL-31 levels in the circulation and in the skin, and these levels decrease when itching improves during treatment, further highlighting that malignant T cells are the source of IL-31 [65]. Similarly, the expression of the IL-31 receptor was increased in the skin of CTCL patients suffering from pruritus compared to asymptomatic patients while receptor expression was not related to disease stage [66]. From the clinical point of view, cases of SS typically show erythroderma and refractory pruritus with disease progression that increasingly affects the ability to sleep and to perform daily activities, leading to a severe impairment in the quality of life [67]. To this regard, available lymphoma treatments with an immunomodulatory action are effective in improving pruritus in these patients and have been shown to concurrently reduce IL-31: traditional treatments include corticosteroids, interferons, bexarotene, and photopheresis, while targeted therapeutics consist in histone deacetylase inhibitors and in the anti-CCR4 monoclonal antibody mogamulizumab [65]. The latter eliminates malignant T cells through a mechanism of antibody-dependent cell-mediated cytotoxicity and therefore directly reduces the number of cells that produce IL-31.

Apart from the association with pruritus, IL-31 is also thought to contribute to immune evasion in HL and in CTCL: in the tumor microenvironment, high IL-31 may shift the balance from anti-tumor Th1 responses—mediated by IFNgamma—towards immunosuppressive Th2 cytokines [68]. In this view, therapeutics that directly target IL-31 in CTCL could prove useful beyond the management of pruritus and help overcome disease-related immunosuppression to fire up anti-lymphoma cell-mediated responses.

Nemolizumab is a humanized monoclonal antibody blocking IL-31 receptor A that is currently approved for the indication of AD-associated pruritus in subjects over 13 years of age in Japan [69] and that has completed a phase 3 trial in prurigo nodularis, a chronic neuroimmune skin condition that is characterized by severe pruritus [70]. In the latter trial, nemolizumab was administered as a subcutaneous injection of 60 mg followed by a monthly injection of 30 or 60 mg according to weight showing a significant reduction in the symptoms and signs associated with the disease. Considering these promising results, the role of this drug could be explored in CTCL as adjunctive treatment addressing disease-associated pruritus and potentially attenuating the established Th2 bias in the lymphoma microenvironment that hampers the development of anti-tumor immunity.

Finally, vixarelimab is an investigative human monoclonal antibody targeting the oncostatin M receptor beta, which is able to block the IL-31 and the oncostatin M pathways. Positive results of a phase 2a trial in prurigo nodularis are currently available [71] and may encourage future investigation in diverse cutaneous conditions characterized by pruritus.

## 6. Targeting Thymic Stromal Lymphopoietin in Cutaneous T Cell Lymphoma

The cytokine TSLP is a potent stimulator of IL-4 and IL-13: in cutaneous inflammatory conditions such as AD, keratinocytes are the main source of TSLP, the production of which is induced by fibroblast-derived periostin, which in turn is stimulated by IL-4 and IL-13 within a proinflammatory milieu [72]. Moreover, the expression of TSLP and its receptor has been demonstrated in different solid tumors, where it provides a potential mechanism of escape from anti-tumor immunity by polarizing immunity towards a Th2 TME [73]. This concept is complicated by evidence from mouse models of skin carcinogenesis that TSLP might also mediate protective effects by augmenting anticancer immunity [74], hinting at diverse effects of TSLP according to cell types and organs. In the setting of CTCL, TSLP can stimulate CD4+ T cells to express IL-4 and IL-13 both directly, through TSLP receptors on these cells [27], and indirectly, through dendritic cells [75]. A pathogenic role of TSLP in CTCL is experimentally supported by the elevated TSLP levels detected in the lesional skin of MF and also in the circulation of patients with SS [76,77]. Moreover, TSLP is able to trigger the expression of IL-4 and IL-13 and to induce proliferation of CTCL lymphocytes both in vitro and in vivo [76,77].

Tezepelumab is a human monoclonal antibody against TSLP that is currently approved as an adjunctive treatment in asthma [78]; however, this drug was not able to achieve statistically significant results in a phase 2a trial of subjects with AD [79]. Another phase 2 study (NCT04833855) was completed in subjects with chronic spontaneous urticaria but results are not yet available.

While a role of TSLP in inducing a Th2 microenvironment in CTCL and possibly promoting the proliferation of malignant T cells is recognized, clinical evidence supporting the therapeutic targeting of this cytokine is missing. Finally, data from novel TSLP blockers in development are highly anticipated [80].

## 7. Conclusions and Future Perspectives

The IL-4 and IL-13 pathways have a key role in supporting CTCL growth: they stimulate CD4+ skin-homing malignant T cells and contribute to establishing an immunosuppressive TME. Additionally, they fuel the onset of the severe and often treatment-resistant pruritus that is associated with CTCL.

The introduction of existing targeted therapeutics against these pivotal cytokines represents a relatively straightforward enhancement to the current CTCL treatment armamentarium and an aid in achieving disease symptom control. Moreover, there is promise in the development of antipruritic drugs with innovative mechanisms, addressing the neuroinflammatory pathways of pruritus implicated in itching cutaneous conditions and potentially in CTCL-related pruritus [54,81]. Finally, targeting pathways related to Th2 inflammation, that is a hallmark of CTCL, could contribute to our understanding of the pathophysiologic mechanisms underlying cutaneous lymphomagenesis and provide novel strategies for disease control. Emerging evidence suggests a role in CTCL for additional players in Th2 responses, such as the OX40-OX40L immune checkpoint [82], and novel agents are in clinical development, such as the anti-OX40L amlitelimab (NCT05769777) (Table 1). According to the authors’ perspective, future drug development in CTCL should also consider antibody–drug conjugates, capable of selectively delivering cytotoxic molecules towards surface receptors—such as 13Ralpha2—that are highly expressed by malignant skin-homing lymphocytes [83].

## Figures and Tables

**Figure 1 life-14-00245-f001:**
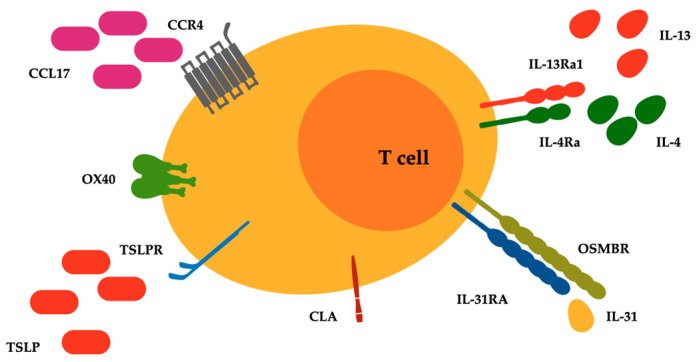
Surface receptors related to type 2 inflammatory pathways expressed on malignant lymphocytes of T cell cutaneous lymphoma. Abbreviations: CCL17, C-C motif chemokine ligand 17; CCR4, C-C chemokine receptor type 4; CLA, cutaneous lymphocyte antigen; IL-4Ra, IL-4 receptor alpha; IL-13Ra1, IL-13 receptor alpha 1; IL-31RA, IL-31 receptor A; OSMRB, oncostatin M receptor beta; OX40, T cell co-stimulatory receptor CD134; TSLP and TSLPR, thymic stromal lymphopoietin and its receptor.

**Table 1 life-14-00245-t001:** Monoclonal antibodies and their antigen targets with effects on the IL-4/IL-13 axis and other related pathways.

Pathway	Drug	Target
IL-4 and IL-13	Dupilumab	IL-4Ra
	Lebrikizumab	IL-13
	Tralokinumab	IL-13
IL-31	BMS-981164	IL-31
	Nemolizumab	IL-31RA
	Vixarelimab	OSMRB
TSLP	Tezepelumab	TSLPR
	MK-8226	TSLP
	Ecleralimab	TSLP
OX40-OX40L	Rocatinlimab	OX40
	Telazorlimab	OX40
	Amlitelimab	OX40L
IL-33	Itepekimab	IL-33
	Etokimab	IL-33

Abbreviations: CCL17, C-C motif chemokine ligand 17; CCR4, C-C chemokine receptor type 4; CLA, cutaneous lymphocyte antigen; IL-4Ra, IL-4 receptor alpha; IL-13Ra1, IL-13 receptor alpha 1; IL-31RA, IL-31 receptor A; OSMRB, oncostatin M receptor beta; OX40 and OX40L, T cell co-stimulatory receptor CD134 and its ligand; TSLP and TSLPR, thymic stromal lymphopoietin and its receptor.

## Data Availability

No new data were created or analyzed in this study. Data sharing is not applicable to this article.

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
