# Peer review of "Role of IL-4 and IL-13 in Cutaneous T Cell Lymphoma"

_life, 2024, doi:10.3390/life14020245_

Round 1

Reviewer 1 Report

Comments and Suggestions for Authors

The authors have written a very clear and indepth manuscript detailing the role of Il4 adn IL-13 in CTCL especially related to their role in pruritus.  They have nicely outlined how these impact pruritus and reported on different targeted therapies that might be effective in itch reduction.  I have minimal concerns with this manuscript.

1. The role of IL-31 in pruritus in CTCL has not been consistently confirmed in the literature. The authors should add a sentence to let the readers know that there is some debate about its correlation with itch in this setting.

Author Response

Dear Reviewer,

Thank you for your careful review of our paper: we have carefully considered each issue raised and modified the manuscript accordingly. Our responses are provided below.

  1. Thank you for your constructive comment: we have added a relevant statement accordingly, supported by a reference from the recent literature. Please see: “However, the extent of the correlation between IL-31 and pruritus is still debated in this setting and a broader role of this cytokine in the pathogenesis of CTCL has been inconsistently confirmed in the current literature [64].”
  2. Olszewska, B.; SokoÅ‚owska-WojdyÅ‚o, M.; Lakomy, J.; Nowicki, R.J. The Ambiguous Pruritogenic Role of Interleukin-31 in Cutaneous T-Cell Lymphomas in Comparison to Atopic Dermatitis: A Review. Postepy Dermatol Alergol 2020, 37, 319–325, doi:10.5114/ada.2020.96260.

Reviewer 2 Report

Comments and Suggestions for Authors

Author Response

Dear Reviewer,

Thank you for your efforts on our paper and for your constructive comments. We have carefully responded to each issue raised and our responses are provided below point-by-point.

  1. Thank you for your suggestion; we have mentioned the aspect of erythema and added a statement on its impact on the quality of life, followed by a relevant citation. Please see: “Skin lesions typically present with erythematous and scaly patches in early stage but gradually form generalized plaques, tumors or erythroderma in the advanced stage [4]. The appearance of erythema is a potential source of severe esthetic impairment and a substantial concern for patients irrespective of subjective symptoms of pruritus [5].”
  2. Thank you, to avoid a potential source of confusion we have changed from “leukemic involvement” to “peripheral blood lymphocytosis”.
  3. Thank you, we have clarified that we are referring to the use of immunomodulators/immunosuppressive medications and therefore changed the term to “exposure to immunomodulating drugs”.
  4. Thank you, we have added the latest evidence from the literature on this topic. Please see: “A 2023 prospective study assessed potential predictive biomarkers in the tear fluid of 39 AD patients initiating dupilumab: ocular adverse events during treatment were significantly associated to low tear break-up time and to high IL-33 levels [40].”
  5. Chiricozzi, A.; Di Nardo, L.; Gori, N.; Antonelli, F.; Pinto, L.; Cuffaro, G.; Piro, G.; Savino, G.; Tortora, G.; Peris, K. Dupilumab-Associated Ocular Adverse Events Are Predicted by Low Tear Break-up Time and Correlate with High IL-33 Tear Concentrations in Patients with Atopic Dermatitis. Exp Dermatol 2023, 32, 1531–1537, doi:10.1111/exd.14859.